# Nutrient Removal from Chinese Coastal Waters by Large-Scale Seaweed Aquaculture Using Artificial Upwelling

**Wei Fan [1], Ruolan Zhao [1], Zhongzhi Yao [1], Canbo Xiao [1], Yiwen Pan [1], Ying Chen [1,2], Nianzhi Jiao [3] and Yao Zhang [3,*]**

1   Ocean College, Zhejiang University, Zhoushan 316000, China
2   The State Key Lab of Fluid Power and Mechatronic System, Zhejiang University, Hangzhou 310027, China
3   State Key Laboratory for Marine Environmental Science and College of Ocean and Earth Sciences, Xiamen University, Xiamen 361101, China
*   Correspondence: yaozhang@xmu.edu.cn

**Abstract:** Ecological engineering by artificial upwelling for enhancing seaweed growth and consequently increasing nutrient removal from seawater has proved promising in combating intense coastal eutrophication. However, a key issue needs to be answered: how much economic and ecological benefit could this engineering bring if it were to be implemented in national aquaculture areas. This study estimated the promoting effect of nutrient concentration change induced by artificial upwelling on kelp growth using a model simulation based on the temperature, light intensity, and nutrient concentration data from three bays in Shandong Province, China— Aoshan Bay, Jiaozhou Bay, and Sanggou Bay. Our results indicate that ecological engineering by artificial upwelling can increase the average yield of kelp by 55 g per plant. Furthermore, based on the current existing kelp aquaculture area of China and the aquaculture density of 12 plants/m$^2$, we inferred that this ecological engineering could increase the natural kelp yield by 291,956 t and the removal of nitrogen (N) and phosphorus (P) nutrients by 4875–6422 t and 730–1080 t, respectively.

**Keywords:** eutrophication; ecological engineering; artificial upwelling; seaweed aquaculture; nutrient removal

## 1. Introduction

In recent years, the marine environment has deteriorated seriously due to the development of industry and agriculture as well as a highly growing population in China [1, 2]. The Chinese eutrophication area was more than 60,000 square kilometers in 2017, of which more than 50% was moderately or severely eutrophic sea areas. Seawater eutrophication can cause a series of hazards including seawater acidification, hypoxia, and frequent occurrence of red tides [3]. Using large-scale seaweed aquaculture to absorb and remove nutrient from seawater is an important way to alleviate seawater eutrophication and realize sustainable development, which can transform excess nutrient into biomass and produce enormous economic value [4, 5]. It is estimated that the nutrient removal from one ha of seaweed aquaculture is equivalent to the N and P annual input from 17.8 hectares and 126.7 hectares of Chinese coastal waters, and the annual removal of N and P by seaweed aquaculture is 75,000 t and 9500 t, respectively [6].

China is one of the world's major seaweed aquaculture countries, and contributed 83%–87% of the global annual production of kelp over the past decade [7]. In the past 30 years, with the development of aquaculture technology, the area and yield of kelp culture in China have steadily

increased. The production of kelp culture in China was 1,486,645 tons in 2017, more than eight times that in 1987. However, with the expansion of the aquaculture area and the increase of the aquaculture density, the water exchange capacity in the aquaculture area was severely weakened and nutrient supplementation was limited, resulting in large-scale disease and death of kelp (Figure 1) [8]. Artificial nutrient supplementation is necessary in order to increase the production of aquaculture. The excess fertilizer and rotted seaweed sink to the seabed, causing nutrient concentrations in sediments to be dozens of times higher than that in the surface layer [9-11]. Because of the weak water exchange capacity in aquaculture areas, the nutrients in the sediments cannot be resuspended effectively to be utilized by kelp and become the endogenous cause of seawater eutrophication.

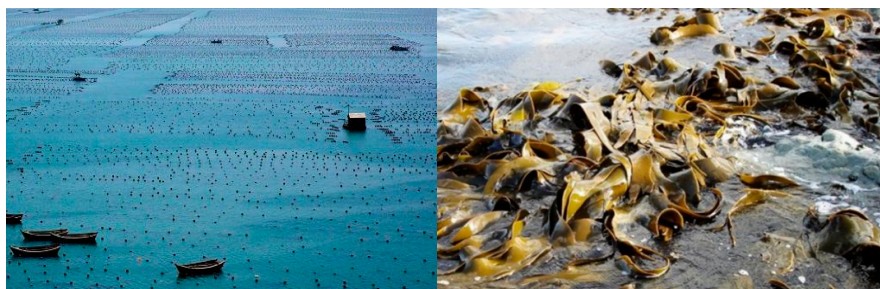

**Figure 1.** High-density kelp aquaculture can readily cause kelp disease.

Artificial upwelling is an important way to orderly release nutrients from sediments and elevate them to the surface layer to promote kelp growth, transform nutrient into biomass, and alleviate endogenous eutrophication of seawater. In the United States, Japan, Norway, and Taiwan (China), several artificial upwelling experiments have been carried out, during which artificial upwelling simulated natural upwelling, elevating nutrient-rich bottom seawater to the surface. Additionally. it was found that the implementation of artificial upwelling system had a good effect on adjusting the N/P ratio [12] and increasing chlorophyll-a concentration in seawater [13, 14]. In order to alleviate the endogenous-nutrient-accumulation problem in Chinese coastal aquaculture areas, an ecological engineering experiment using artificial upwelling was conducted in Aoshan Bay, Shandong Province, China. The preliminary sea trials confirmed that artificial upwelling can promote kelp growth and nutrient removal from seawater. However, the key issue that needs to be answered is how much economic and ecological benefit this engineering can bring when it is implemented in the aquaculture areas.

In this study, we used the model of kelp growth to evaluate the effects of artificial upwelling technology on kelp growth. The Shandong Peninsula is one of the most important kelp aquaculture regions in China. Three semi-closed bays in Shandong Province, Aoshan Bay, Jiaozhou Bay, and Sanggou Bay, are typical aquaculture areas. The nutrient concentration of N and P in the sediments of the three bays is much higher than that on the surface, especially in Sanggou Bay, where the nutrient concentration is 80 times higher than that in the surface layer. We chose the three bays as the study area and analyzed the effects of nutrient concentration changes on kelp growth using the kelp growth model. On this basis, we estimated the economic and ecological benefits of ecological engineering on the existing aquaculture area by assuming that the aquaculture density was 12 plants/m². 

This paper is organized as follows. First, we introduce ecological engineering by artificial upwelling which has been successfully implemented in Aoshan Bay. Next, the simulation process and the results of kelp growth are presented. Finally, the problems and suggestions that should be addressed for the implementation of the engineering are discussed.

## 2. Ecological Engineering by Artificial Upwelling

The ecological engineering is located in Aoshan Bay (120°37' E–121 °00' E, 36°16' N–36°29' N), Shandong Province, China, as shown in Figure 2. It is a typical semi-closed bay with vast shallow coastal waters and has been developing large-scale aquaculture since the 1980s. In recent years,

nutrient deficiency has occurred in the surface seawater, while the nutrient concentration in the sediment is much higher than that in the surface water as shown in Table 1, which hinders the development of mariculture in Aoshan Bay. Therefore, Aoshan Bay is suitable for implementing engineering by artificial upwelling, which releases nutrients from the sediment and lifts them to the surface seawater to promote the development of mariculture. An artificial upwelling system was deployed and the experiment for kelp aquaculture was conducted in Aoshan Bay. As shown in Figure 3, the artificial upwelling system mainly consists of two parts: an offshore solar power generation floating platform and air-injection systems. The floating platform was used to provide power to the air-injection systems to form a large-scale rising bubble plume. The rising bubble plume transports the nutrient-rich bottom water or porewater to the surface, which can enhance the nutrient concentration in the euphotic layer. The detailed structure can be found in Lin et al. (2019) [15]. In this experiment, the total area of kelp aquaculture is 48,000 $m^2$, of which 8,400 $m^2$ are attributable to the artificial upwelling distribution area. A total of 24 air-injection systems were deployed in the artificial upwelling distribution area. As shown in Figure 3, the air-injection tube was fixed with the kelp mariculture raft from the solar platform to the seabed at 10 m depth. The experiment began in November 2018. The air-injection systems worked continuously for 2 h at noon every day in the artificial upwelling distribution area, while no fertilization measures were taken in the rest of the area. In such a controlled experiment, the growth of kelp is only related to whether the artificial upwelling system is implemented or not.

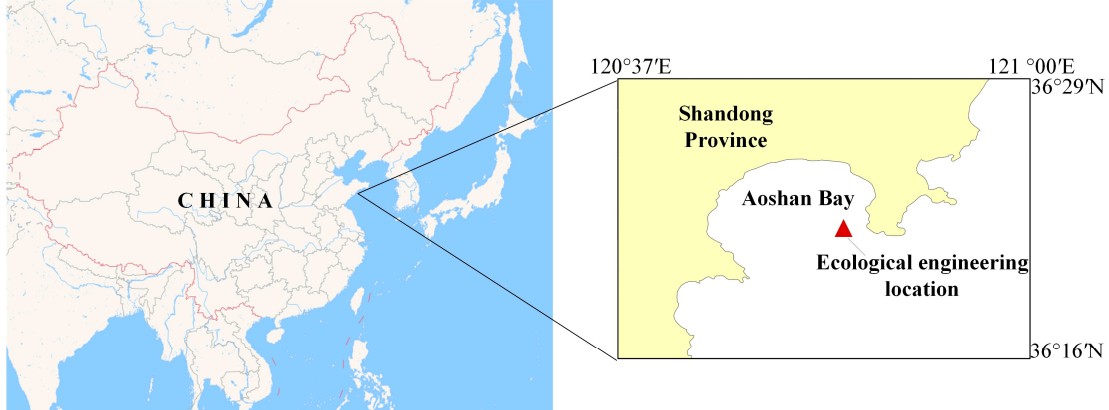

**Figure 2.** The location of ecological engineering in Aoshan Bay.

**Table 1.** Nutrient concentration in Aoshan Bay.

|  | Nitrogen ($NO_3^-+NO_2^-$) | Phosphorus ($PO_4^{3-}$) |
|---|---|---|
| Surface water (µmol/L) | 1.87 | 0.36 |
| Sediment (µmol/L) | 101.82 | 2.20 |

According to the distance from the upwelling system, the aquaculture area was divided into three regions. Region 1 is the distribution area of the artificial upwelling system, region 2 is the area near the artificial upwelling system, and region 3 is the area far from the artificial upwelling system (Figure 4). By 1 March 2019, kelp had been cultured for 3–4 months. We harvested some kelp and measured the weight to determine the promotion effect of artificial upwelling on kelp growth. In the measurement, three groups of kelp were harvested, each of which contained 30 kelp plants from three different regions (ten plants every region). The harvested kelp was weighed after being dried on the shore, so the result can be regarded as dry weight. As shown in Table 2, the average weight per plant is 21.2 g in the upwelling area, 46.2 g in the area near the upwelling system, and 10.1 g in the area far from the upwelling system.

Contrary to expectations, the area with the largest weight of kelp is not where the upwelling system distributed but rather the area near the upwelling system. The reason for this phenomenon is

the influence of tidal current in the aquaculture area. Under the action of tidal current, the upwelling plume will flow with the tidal current to the area near the upwelling distribution area, namely the area of upwelling most influenced is the surrounding area of the upwelling distribution area. The area far away from the upwelling distribution area are seldom affected by upwelling, even in the case of tidal current, so the average weight of kelp is the smallest.

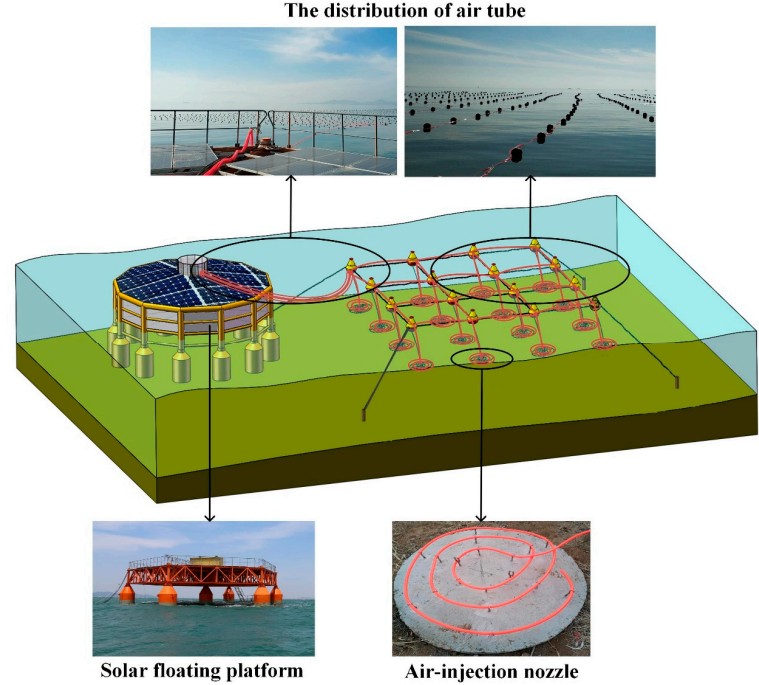

**Figure 3.** Schematic diagram of artificial upwelling system.

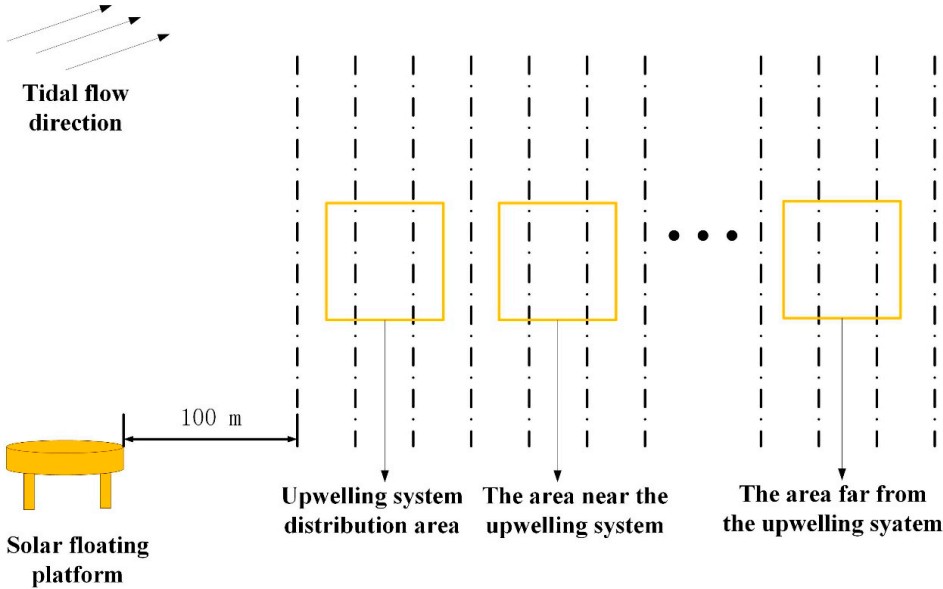

**Figure 4.** Schematic diagram of artificial upwelling system distribution.

**Table 2.** Kelp weight in different regions.

| | Kelp weight (g) | | |
|---|---|---|---|
| Group | Upwelling area | Near the upwelling area | Away from upwelling area |

| | | | | | |
|---|---|---|---|---|---|
| | 11.3 | 28.7 | 44 | 27.8 | 10.3 | 8.6 |
| | 15.8 | 12.6 | 37.6 | 66.3 | 12.5 | 8 |
| 1 | 13.4 | 23.1 | 62.1 | 40.9 | 11 | 6.2 |
| | 13.1 | 22.4 | 44.1 | 46.8 | 8.2 | 6.1 |
| | 18.8 | 10.8 | 62.9 | 40.5 | 9.9 | 5.2 |
| | 16.9 | 56.8 | 63.2 | 71.7 | 17.7 | 12 |
| | 24.8 | 28.7 | 32 | 53.1 | 15.1 | 6.9 |
| 2 | 29.5 | 23 | 72 | 87.5 | 10 | 9 |
| | 19.5 | 14.8 | 79.8 | 36.8 | 13.8 | 7.4 |
| | 25 | 78.8 | 44.9 | 50.7 | 14.8 | 4.7 |
| | 25.6 | 14.6 | 34.6 | 32.3 | 7.7 | 6 |
| | 15.9 | 18.1 | 38.4 | 36.9 | 12.2 | 11 |
| 3 | 8 | 10.3 | 37.6 | 42.9 | 14.3 | 10.2 |
| | 9.1 | 13.8 | 19.1 | 23.7 | 13.2 | 10 |
| | 21 | 11.1 | 24 | 32.5 | 12.6 | 9 |
| Average weight of per plant | 21.2 | | 46.2 | | 10.1 | |

## 3. Material and Method

Sea trials have proved that engineering by artificial upwelling has the potential of assisting kelp growth and alleviating the endogenous-nutrient-accumulation problem. However, if the engineering is deployed in the total aquaculture area of China, how much kelp will be produced and how much nutrient will be removed are unknown. In this section, we chose the three bays in Shandong Province as the study area, using the model of kelp growth to simulate the economic and ecological effects of the engineering.

### 3.1. Study Area

Shangdong Province (34°22′ N–38°24′ N, 114°19′ E–122°43′ E) is a coastal province in China. Shangdong Peninsula has more than 3000 km of coastline, accounting for 1/6 of the national coastline. It is a major province of seaweed aquaculture in China with seaweed production and aquaculture area accounting for 25%–60% of the national seaweed production and aquaculture area from 1979– 2017 (Figure 5). Therefore, the study of the implementation of an artificial upwelling system in Shandong Province has significance for the national implementation of the engineering. For this purpose, we chose three semi-closed bays in Shandong Province, which are Sanggou Bay, Aoshan Bay. and Jiaozhou Bay, as the research sites (Figure 6). Aoshan Bay, located in Qingdao City, Shandong Province, was the site of our ecological engineering demonstration. Sanggou Bay and Jiaozhou Bay, which are the two famous kelp farms in Shandong Province, are located in Weihai City and Qingdao City, respectively. As shown in Table 3, the tidal currents of the three bays are all regular semi-diurnal tides, with two fluctuations each day. The average tidal current velocity in Aoshan Bay is 25 cm/s, and the maximum velocity is 85 cm/s in the southeastern estuary [16]. The hydrodynamic condition and water exchange capacity of Sanggou Bay and Jiaozhou Bay are slightly better than that of Aoshan Bay. The average tidal current velocities are 35 cm/s and 45 cm/s [17], respectively. Kelp aquaculture needs a sea area with a smaller wave but better water exchange ability. The research shows that the sea area with a velocity of 80 cm/s is the most suitable area for seaweed aquaculture [18]. The three bays have relatively low tidal velocity, which is suitable for the deployment of kelp rafts and artificial upwelling systems. At the same time, the artificial upwelling system can make up for the weak hydrodynamic exchange. However, hydrological conditions and eutrophication levels

of the three bays are different due to the different geographical environments. It is necessary to study the effects of the artificial upwelling system on the kelp aquaculture of the three bays.

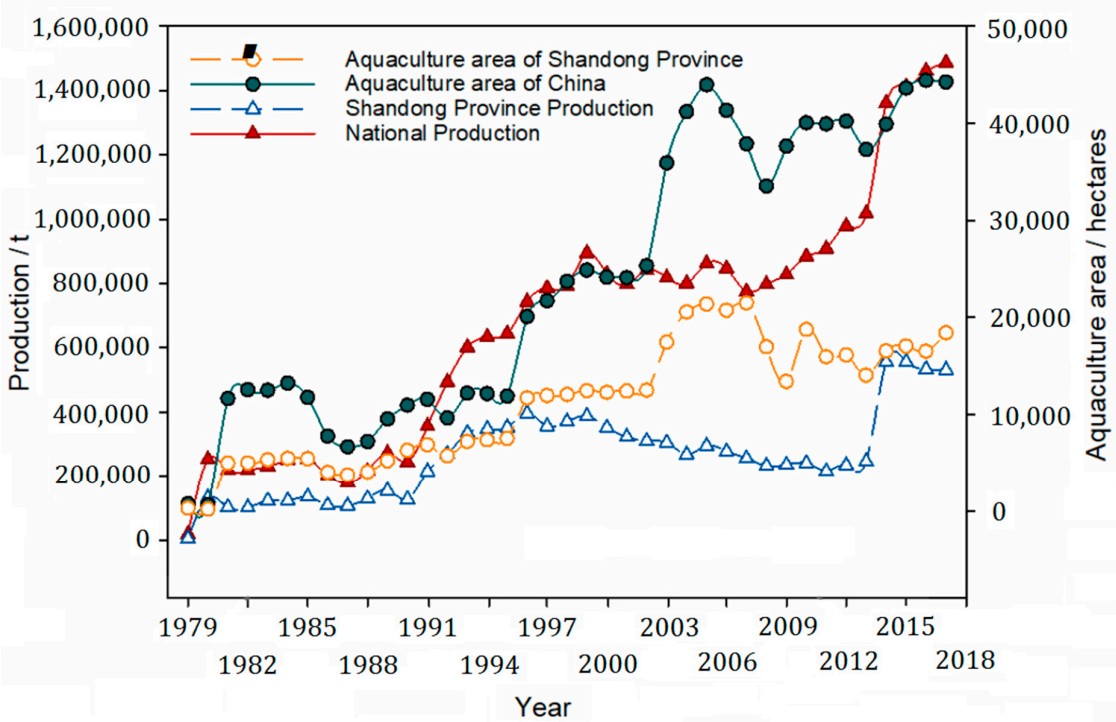

**Figure 5.** Kelp annual yield and aquaculture area in Shandong Province and the whole country from 1979 to 2017. Data are from the China Fisheries Statistics Yearbook (1979–2017).

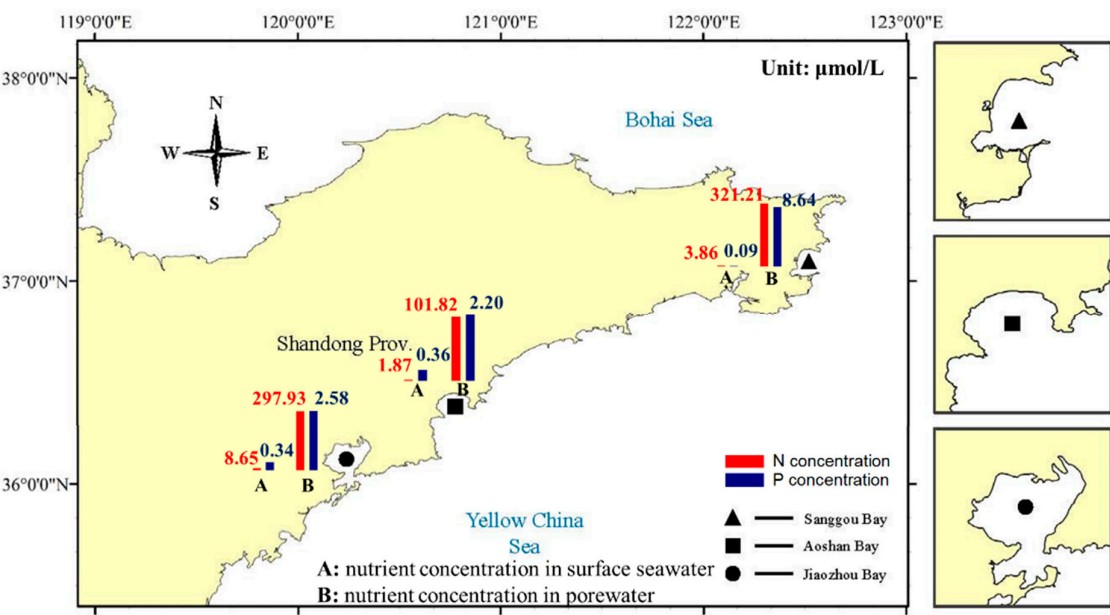

**Figure 6.** Nutrient distribution in the study area.

**Table 3.** Hydrologic condition of the three bays.

| Sea area (km²) | Average water depth (m) | Tidal current type | Average velocity (cm/s) |
| --- | --- | --- | --- |

| Aoshan Bay | 164 | 4 | Regular semi-diurnal tides | 25 |
|---|---|---|---|---|
| Sanggou Bay | 150.3 | 7.5 | Regular semi-diurnal tides | 35 |
| Jiaozhou Bay | 500 | 7 | Regular semi-diurnal tides | 45 |

In recent years, in the process of aquaculture, the accumulation of residual fertilizer, decayed seaweeds, and fish feces has led to the increase of organics and nutrient in the sediment. However, the nutrient concentration in the surface seawater remains at a low level, which makes it difficult to maintain the nutrient requirement during the seaweed growing season. Taking Sanggou Bay as an example, the DIN (dissolved inorganic nitrogen including $NO_3^-$ and $NO_2^-$) and $PO_4^{3-}$ concentration of the surface seawater are 3.86 µmol/L and 0.09 µmol/L [19], respectively, while in the sediment they are 321.21 µmol/L and 8.64 µmol/L, respectively. The ratio of nutrient concentration in the sediment to that in the surface water reaches 80–100. The same situation is also found in Jiaozhou Bay and Aoshan Bay (Figure 6). It is inferred that the endogenous-nutrient-accumulation problem may be a common phenomenon in the semi-closed bays of Shandong Province.

*3.2. Dynamic Individual Growth Model of Mariculture Kelp*

Kelp is an autotrophic plant, which can accumulate organic compounds by photosynthesis. There are many factors of photosynthesis such as light, seawater temperature, nitrogen and phosphorus concentrations, etc. The light and seawater temperature are uncontrollable factors in the process of kelp aquaculture. However, the nutrient concentration could be changed manually. Hence, the purpose of the engineering is to increase the nutrient concentration in the surface seawater, thereby promoting the growth of kelp and increasing the amount of nutrient removal. In order to estimate the economic and ecological benefits of the artificial upwelling project, we should first understand the mechanism of the nutrient's effect on the growth of kelp.

The effects of light, temperature, and nutrient concentration on the growth of kelp are dynamic and complex. Wu et al. established a mathematical model to simulate and predict the growth process of kelp based on the experimental data of the physiology and ecology of kelp and the observed data of light intensity, temperature, nutrient concentration, and other environmental parameters affecting the kelp growth [20, 21]. In this model, the net growth ($N_{growth}$) of kelp is determined by its total growth ($G_{growth}$) and respiration (*resp*):

$$N_{growth} = G_{growth} \times (1 - resp) \tag{1}$$

where the *resp* is mainly controlled by water temperature [22]:

$$resp = \frac{0.3197 \times T^2 - 6.5728 \times T + 52.851}{100} \tag{2}$$

$G_{growth}$ is determined by maximum growth rate ($\mu_{max}$), photosynthetic effective radiation (*I*), temperature (*T*), and nutrient content in kelp (*NP*) [23]:

$$G_{growth} = \mu_{max} \times f(I) \times f(T) \times f(NP) \tag{3}$$

where *f(I)* is the effective photosynthetic irradiation, which is one of the main factors affecting kelp yield. The effective photosynthetic irradiation is affected by factors such as the depth of kelp aquaculture and the turbidity of seawater, which can be expressed by the following formula:

$$f(I) = \frac{I}{I_0} exp(1 - \frac{I}{I_0}) \tag{4}$$

where the $I_0$ (W/m²) is the best light intensity for kelp growth, *I* (W/m²) is the light intensity of the underwater at the aquaculture depth after attenuation of the water, according to the Lambert–Beer law:

$$I = I_s e^{-kz} \tag{5}$$

where $z$ (m) is the depth of kelp aquaculture, $I_s$ (W/m²) is the light intensity on the surface of the seawater and $k$ is the attenuation coefficient of light, which is affected by suspended particles in water. It can be obtained by following the empirical formulas:

$$k = 0.0484 \, \text{TPM} + 0.0243 \tag{6}$$

where TPM (mg/L) is the total suspended solids concentration.

In Formula (3), the effect of sea water temperature on kelp is as follows:

$$f(T) = exp(-2.3(\frac{T - T_0}{T_x - T_0})) \tag{7}$$

where $T$ is the temperature of seawater, $T_0$ is the optimum temperature for kelp growth, and $T_x$ is the temperature limit for kelp growth.

Kelp absorbs a different ratio of nitrogen to phosphorous during its life cycle because the photosynthesis of kelp is directly affected by free N and P in kelp tissues rather than nutrients in external waters. Therefore, the nutrient factors influencing kelp growth are determined by the contents of N and P in tissues, which can be expressed as follows:

$$f(NP) = Min(f(N), f(P)) \tag{8}$$

where the $f(N)$ and $f(P)$ are the factors influencing nutrient N and P, respectively, including free N and P in kelp tissues and nutrient concentration in seawater. Free N and P in kelp tissues affect the absorption rate of nutrients in seawater, which are not explained in detail in this paper and interested readers can refer to Wu et al. (2009) [20].

In this paper, the model is used to calculate the dry weight change of cultured kelp and the amount of nutrient removal from sediment after the implementation of the artificial upwelling system.

### 3.3. Data Acquisition

Because the growth period of kelp in this model is 180 days, generally from November to May of the next year, we need to obtain the data of light intensity, temperature, and nutrient concentration in this period.

We obtained data on light and temperature from NASA Prediction of Worldwide Energy Resources [24](https://power.larc.nasa.gov/). Figure 7 shows a comparison of light intensity and temperature among the three bays from 15 November 2016 to 13 May 2017. The light intensity of the three bays is generally similar. Especially, the light intensity of Aoshan Bay and Jiaozhou Bay are the same because of their similar geographical location. The temperature of the three bays has similar trends, however, the range of temperature variation in Jiaozhou Bay is significantly higher than that in the other two bays (Figure 7b).

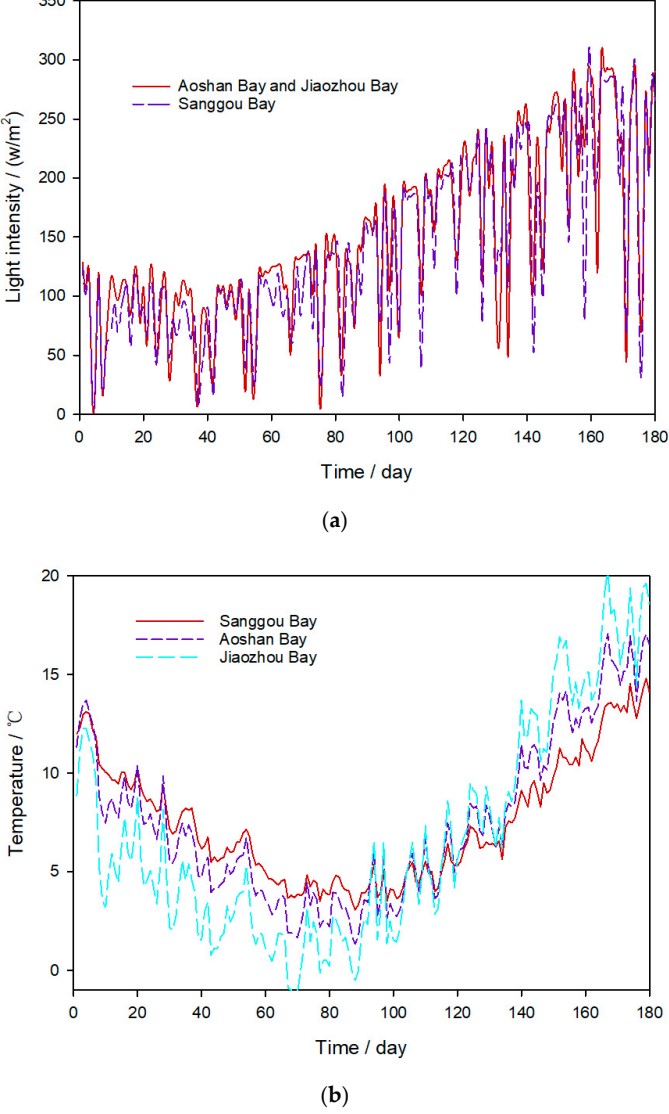

**Figure 7.** Light intensity (**a**) and temperature (**b**) in Aoshan Bay, Sanggou Bay, and Jiaozhou Bay.

The daily data of nutrient concentration is difficult to obtain. We sampled the water of Aoshan Bay in November 2018, March 2019, and May 2019, and obtained the nutrient concentration data of the three seasons in Aoshan Bay (Table 4). The nutrient concentration data of Sanggou Bay and Jiaozhou Bay are derived from the literature (Table 4), which reported the surface nutrients

concentrations in three different seasons. The data in winter was used as the background value for the first 60 days of kelp growth period, the data in spring as the background value for the middle 60 days, and the data in summer as the background value for the last 60 days.

**Table 4.** Background nutrient concentration of the three bays. (DIN = dissolved inorganic nitrogen)

| Bay | Nutrient concentration (μmol/L) | | | | | | Year | Reference |
|---|---|---|---|---|---|---|---|---|
| | Winter | | Spring | | Summer | | | |
| | DIN | $PO_4^{3-}$ | DIN | $PO_4^{3-}$ | DIN | $PO_4^{3-}$ | | |
| Aoshan Bay | 11.86 | 0.82 | 8.96 | 0.29 | 3.16 | 0.26 | 2018–2019 | This study |
| Sanggou Bay | 9.28 | 0.30 | 13.03 | 0.13 | 10.12 | 0.08 | 2010 | Zhang et al. [25] |
| Jiaozhou Bay | 26.42 | 0.35 | 25.00 | 0.19 | 20.00 | 0.19 | 2014 | Gao et al. [26] |

## 4. Results

### 4.1. Simulated Results of the Kelp Growth Model

In order to research the effect of nutrient concentration on kelp growth, the nutrient concentration was constantly changed to calculate the model. Different nutrient concentrations correspond to different growth curves and the final dry weight of kelp. On the basis of the original nutrient concentration, the N and P concentrations were increased to 1.1, 1.2, 1.3, ···, 2.8, 2.9, and 3.0 times the original nutrient concentration, respectively, to simulate the new kelp growth curve.

As shown in Figure 8, when the nutrient concentration is at the initial value, the dry weight per plant in Sanggou Bay and Aoshan Bay can reach to 183 g and 131 g, respectively, while it is only 50 g in Jiaozhou Bay. The main factors restricting the growth of kelp in Jiaozhou Bay are as follows: 1) The growth of kelp is limited by the relative lack of nutrient. Nutrients are most easily absorbed by kelp when the ratio of N to P is 12–16, but the ratio in Jiaozhou Bay is 74–130, which is phosphorus limited. 2) Kelp is a kind of fast-growing plant and very sensitive to temperature. In an environment with a temperature change of 1–2 ℃, the growth of kelp will be different, or even have huge discrepancy. Kelp can grow 10–13 cm longer each day at a temperature of 6–8 ℃, but it will stop growing at 0 ℃, and decay at 20 ℃ or above [20, 21, 27]. As shown in Figure 7b, the temperature in Jiaozhou Bay is generally 1–2 ℃ lower than that in Aoshan Bay and Sanggou Bay. The temperature in Jiaozhou Bay is below 0 ℃ for a few days and nearly 20 ℃ for some days, while the other two bays do not have such conditions.

When the nutrient concentration rose to triple the initial value, the kelp growth curves of Aoshan Bay and Sanggou Bay changed significantly, and the kelp growth curve of Jiaozhou Bay changed slightly (Figure 8). In this process, the production of kelp increased by 30 g per plant in Aoshan Bay and 80 g in Sanggou Bay, but only 3 g in Jiaozhou Bay. This indicates that the main factors restricting the growth of kelp in Jiaozhou Bay are not the concentrations of nutrients, but the N/P ratio and temperature.

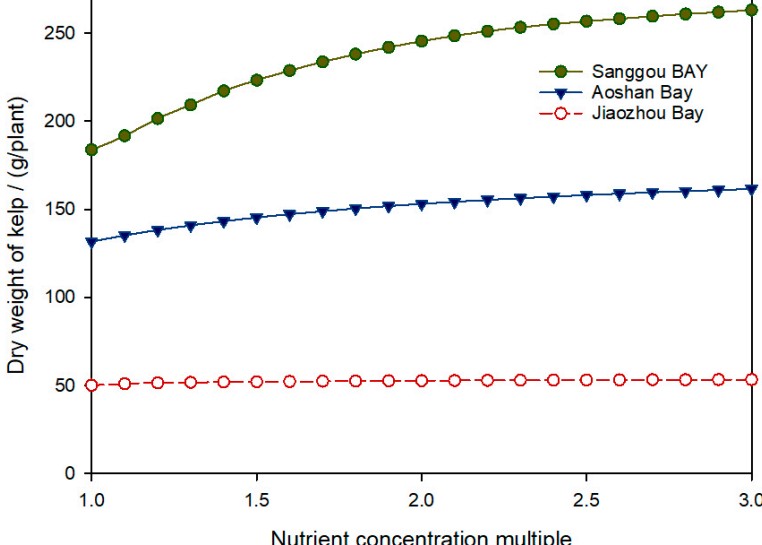

**Figure 8.** Curves of final dry weight of kelp varying with the   nutrient concentration multiple.

In addition, it can be seen that in a certain range, the increase of nutrient can promote the growth of kelp, but with a continuous increase of nutrient concentration, the promotion gradually decreases (Figure 8). Although the trend is similar, the effects of nutrient concentration change on the growth of kelp in the three bays are different. The increase of nutrient concentration has little effect on the growth of kelp in Jiaozhou Bay, and double nutrient concentration can only increase the dry weight of kelp by 1.5 g. However, it has a significant impact on the kelp growth in Aoshan Bay and Sanggou Bay. Especially in Sanggou Bay, medium-yield kelp can grow into high-yield after nutrient upwelling.

*4.2. The Effect of Kelp Growth and Nutrient Removal in the Natural Aquaculture Area*

From Figure 8, if the surface nutrient concentration can be enhanced to three times the original, the yield of each kelp can be increased by 80 g in Sanggou Bay and 30 g in Aoshan Bay. Nitrogen and phosphorus account for 1.67%–2.2% and 0.25%–0.37% of kelp dry weight, respectively [20, 28]. Therefore, the engineering can promote 1.34–1.76 g removal of N and 0.20–0.29 g removal of P per plant in Sanggou Bay and 0.50–0.66 g removal of N and 0.07–0.11 g removal of P per plant in Aoshan Bay. If this ecological engineering is successfully implemented in all kelp culture areas in China, the economic and ecological benefits are estimated as shown in Table 5, where the increase in dry weight of each kelp is the average of the values in Aoshan Bay and Sanggou Bay, and the aquaculture area of Shandong Province and China is from the China Fishery Statistical Yearbook (2018). Thus, the engineering can increase the production of kelp by 121,419 t/year in Shandong Province and 291,956 t/year in the whole of China (Table 5), which is almost 19.6% of the annual kelp production in China. At the same time, the N and P nutrient removal increased by this engineering is considerable (2028– 2671   t  N and 303–449 t P in Shandong Province and 4875–6422 t N and 730–1080 t P in the whole of China) (Table 5).

**Table 5.** Increased yield and nutrient removal of kelp aquaculture in Shandong Province and China.

| Items | Data | Units |
|---|---|---|
| Increased dry weight | 55 | g/plant |
| Aquaculture area in Shandong | 18,397 | Hectares in 2017 |
| Aquaculture area in China | 44,236 | Hectares in 2017 |
| Culture density | 12 | Plant/m$^2$ |
| Increased dry weight in Shandong | 121,419 | t/year |
| Increased dry weight in China | 291,956 | t/year |
| N concentration | 1.67–2.2 | % Dry weight |
| P concentration | 0.25–0.37 | % Dry weight |
| Increased N removal in Shandong | 2028–2671 | t/year |
| Increased N removal in China | 4875–6422 | t/year |
| Increased P removal in Shandong | 303–449 | t/year |
| Increased P removal in China | 730–1080 | t/year |

Increased kelp production can be of considerable economic value, and the additional removal of N and P can also alleviate the endogenous-nutrient-accumulation problem and improve the marine environment. Besides, considering that a large amount of fertilizer is put into kelp aquaculture area every year, the implementation of this project not only saves the economic expenditure of this part but also avoids a great deal of exogenous nutrient input.

## 5. Discussion and Conclusion

### 5.1. Engineering Feasibility

According to the simulation results, taking Sanggou Bay as an example, if the surface nutrient concentration can be increased to 2–3 times the initial concentration, it will bring greater economic and ecological benefits.

Taking Sanggou Bay as an example, this paper calculated the nutrient concentration in the surface seawater in order to estimate how many air-injection systems are needed for every 1000 m$^2$ of aquaculture area, and how many hours it takes to increase the nutrient concentration to three times the original surface concentration. Assuming that the area of the aquaculturea area is 1000 m$^2$, and the depth of the aquaculture area is 3 m, the nutrient concentration at the surface of the aquaculture area is as follows:

$$c = \frac{c_1(1000 \times 1000 \times 3 - 60 \times nQ_Wt) + 60 \times c_2 nQ_Wt}{1000 \times 1000 \times 3} \tag{9}$$

where $c_1$ (μmol/L) is the concentration of nutrient in the surface layer; $c_2$ (μmol/L) is the concentration of nutrient in the bottom water when it is raised to the surface layer by artificial upwelling, n and $t$ (*h*) are the number of air-injection systems and their working hours, respectively, $Q_W$ (L/min) is the flow rate of artificial upwelling. There is a proportional relationship between flow rate ($Q_W$) and air injection rate ($Q_0$): $Q_W = (80–100) Q_0$. Here the $Q_0$ of each air-injection system is 100 L/min. In this case, according to the theoretical model of artificial upwelling proposed by Liang, we can roughly calculate that the ratio of nutrient concentration of plume to that of sediment is about 1:18 when the artificial upwelling project is carried out in a sea area with depth of about 10 m [29].

In fact, the weight of phosphorus is heavier than that of nitrogen so that the uplifting amount of these ions by the air bubble is different for each. However, in this study, due to the large volume of flow rate induced by air-injection, we did not consider this problem and only used Formula 9 to calculate the lifting of N, P nutrient. In the process of actual engineering implementation, the mechanism of lifting nutrient from the seabed to surface results from the combination of the suspension and falling velocity of matter associated with water turbidity. Further work will have to determine the influence of the water turbidity, tidal current velocity, and other factors on the air-lift

efficiency. Actually, we have done some preliminary research in this area, the detailed information can be found in Lin et al. (2019) [15].

The DIN and $PO_4^{3-}$ concentration of surface seawater in Sanggou Bay are 3.86 μmol/L and 0.09 μmol/L, respectively, and that of sediment are 321.21 μmol/L and 8.64 μmol/L, respectively. Based on these data, we can derive the expression of c from Formula 9, whose independent variables are n and t. Let $c: c_1 = 3$ (increase the surface nutrient concentration to three times its original nutrient concentration), the relationship between the working time and the number of air-injection systems can be obtained (Figure 9).

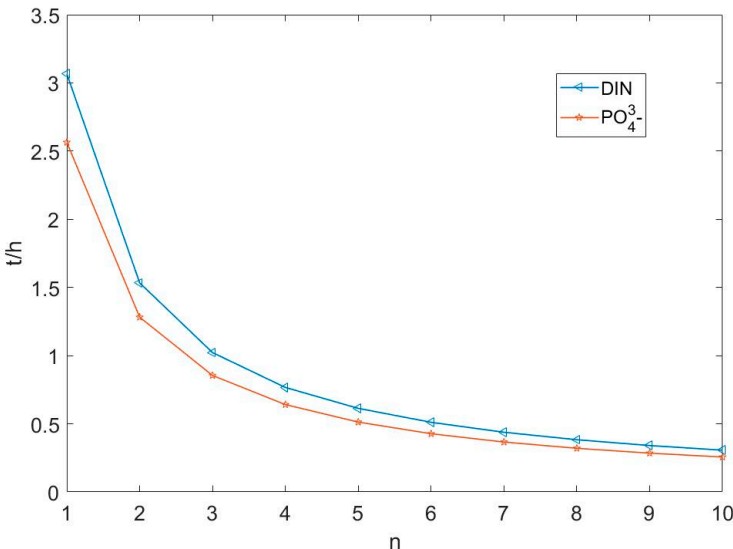

**Figure 9.** The relationship between the working time and the number of air-injection systems.

When only one air-injection system is placed in every 1000 m² of aquaculture, after 2.56 h of continuous working of the air-injection system, the surface $PO_4^{3-}$ concentration can reach three times the original nutrient concentration, but it needs 3.07 h to make the surface DIN concentration meet the requirements. When two air-injection systems are deployed, the time required is 1.28 and 1.53 h, respectively. As the number of air-injection systems increases, the time difference between them decreases gradually. However, considering energy consumption, a suitable number of air-injection systems is required. According to the above results, it is suggested that 1–2 air-injection systems should be deployed in every 1000 m² of aquaculture area.

The air-injection system with a $Q_0$ of 100 L/min consumes 1.1 kW·h per hour. If an air-injection system is deployed in every 1000 m² of sea area and the working time is 3.5 h per day when the growth rate of kelp is the fastest, then a solar energy platform can be used to carry out artificial upwelling project in 15,000 m² of aquaculture sea area. Therefore, in theory, it is feasible to increase the nutrient removal effect of kelp culture through the implementation of artificial upwelling engineering.

*5.2. Four Suggestions on Engineering Implementation*

The purpose of this study was to assess whether artificial upwelling can promote kelp culture and further alleviate the problem of endogenous-nutrient-accumulation in coastal areas of China. After the analysis of sea trial data and the calculation of the model for kelp growth, it was found that the artificial upwelling project can effectively promote the growth of kelp in aquaculture areas where the accumulation of endogenous nutrient is serious. Implementing artificial upwelling engineering in a suitable aquaculture area will bring great economic and ecological benefits. However, there are still several factors needed to be considered in order to successfully implement the engineering in the Chinese coastal area and achieve a better effect.

1) Select suitable aquaculture areas to implement ecological engineering.

From the results of Figure 8, it can be seen that not all sea areas are suitable for kelp aquaculture, nor all sea areas are suitable for the implementation of engineering. The results of the three sea areas with the same endogenous-nutrient-accumulation problem are quite different. Therefore, when choosing the sea area for project implementation, we should try to choose a sea area similar to Sanggou Bay, and avoid choosing a sea area like Jiaozhou Bay. That is to say, on the major premise of having a serious problem of endogenous-nutrient-accumulation, choose the sea area where the natural environment is suitable for kelp growth. For example, kelp aquaculture is suitable for waters with less wind and waves, smooth tidal current, and high transparency. Furthermore, we should choose the sea area where the nutrient concentration in the surface layer is the main limiting factor for kelp aquaculture. Only in this way can the engineering exert the optimum benefits.

2) Using different implementation schemes in different sea areas.

The implementation schemes mentioned here include two aspects: a) Target concentration setting for the surface nutrient elevation; b) Combination mode of the number of air injection systems and working time. Taking Sanggou Bay as an example, the dry weight of kelp is only 1.3 g different when the surface nutrient is raised to twice or triple the original level, but there is a great difference in the gas injection time. According to calculation, it takes only half of the time to double the original nutrient as compared with tripling the original nutrient when one air injection system is placed in every 1000 m$^2$. Then, in the case of a certain amount of electricity being provided by the solar platform, the aquaculture area can be increased and the efficiency can be improved.

For different bays, because the ratio of nutrient concentration in porewater to that in surface water is not the same, even if the nutrient concentration increased by the same multiple, the number of air injection systems per 1000 m$^2$ and the insufflation time are quite different. Therefore, according to the specific situation of each bay, we should choose the most suitable combination of the number of air injection systems and working time for the bay. Moreover, the demand for nutrient in kelp is different at different stages of its growth, which can be used to determine the working time of artificial upwelling.

3) Continuously optimize engineering and reduce costs.

At present, compared with other artificial upwelling technologies, the air-injection artificial upwelling device we have studied has lower cost and higher efficiency. However, due to the difficulty of obtaining energy on the ocean, it needs a higher cost to build a large-scale solar platform and there are more maintenance costs. Besides, although the power supply can provide a large area for the implementation of the engineering, it needs to distribute a large number of air tubes, which not only increases the difficulty of construction but also reduces the air injection rate.

Therefore, in future research, the artificial upwelling system should be continuously optimized to achieve the goal of low cost, easy deployment, and high air injection efficiency.

4) Combining with other engineering methods to further promote kelp growth.

In addition to nutrient limitations, kelp growth is also limited by other factors, such as the temperature in Jiaozhou Bay. Similarly, in some sea areas, due to the high turbidity of seawater, the lack of light will also hinder the growth of kelp. Besides, the kelp will be damaged by the waves or drift away in some sea areas with strong wind and waves, which will cause huge losses [30-32]. In these cases, nutrient supplement and wave absorbing devices can further promote the growth of kelp. If the artificial upwelling engineering is combined with these methods, it can not only increase the kelp production and nutrient removal but also expand the application scope of the engineering and solve the endogenous-nutrient-accumulation problem in a larger area of the Chinese coastal waters.

Nowadays, with the deterioration of the ocean environment, the seaweed aquaculture industry is also facing great challenges. The contradiction between the deterioration of the ocean environment and the large-scale commercial marine fertilization and aquaculture production has always existed. Hence, we urgently need a solution to ensure aquaculture production without polluting the marine environment. Artificial upwelling not only can lift nutrient-rich seawater from the bottom to the surface, meet the nutrient demand of kelp growth, and ensure its yield, but also utilize excess nutrient in porewater to alleviate the accumulation of endogenous nutrient and improve the marine

environment. This method shows an opportunity to slow down or even stop the deterioration process of the Chinese coastal environment.

**Author Contributions:** Conceptualization, Y.Z. and W.F.; methodology, software, validation and writing—original draft preparation, W.F, and R.Z.; investigation and formal analysis, Z.Y., and C.X.; writing—review and editing, Z.Y. and W.F.; supervision, W.F., Y.P.,   Y.C., and N.J..

**Funding:** This research was funded by the National Key Research and Development Program of China (No. 2016YFA0601400) and the National Natural Science Funds of China (No. U1805242 and No. 41976199).

**Conflicts of Interest:** The authors declare no conflict of interest.

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
