# Peer review of "Nutrient Removal from Chinese Coastal Waters by Large-Scale Seaweed Aquaculture Using Artificial Upwelling"

_water, doi:10.3390/w11091754_

Round 1
Reviewer 1 Report
1. Different shape of bay area has its own speed, direction, occurrence and type of current. These characteristics of current in each bay area significantly affect the nutrient dispersion, kelp growth and the effect of artificial upwelling system, respectively. This paper will be better if author collects and analyzes the relevant data about three bay areas.
2. Kelp absorbs different ratio of nitrogen to phosphorous during its life cycle. The analysis of this paper has to consider the amount and season of fertilization.
3. The transparency of seawater also affects the photosynthesis activity of kelp. Besides considering the effect of light and temperature, the transparency of seawater is another important factor during the process of photosynthesis.
4. Ions of nitrogen and phosphorous are main elements of eutrophication, the data cited from 1996 to 2014 in table 3 is inconvenient for strongly developing China.
5. The weight of phosphorous is heavier than that of nitrogen so that the uplifting amount of these ions by air bubble is different from each other. Calculating nutrient removal has to think about this factor.
6. The mechanism of lifting nutrient from seabed to surface results from the combination of the suspension, falling velocity of matter associated with water turbidity. From this aspect, the author could objectively justify the artificial upwelling system is the best way for nutrient removal.
7. Please use “m2” instead of “m2” in Lines 21, 73 and 334.
8. Please use “hectare” instead of “ha” in Line 36.
9. Please use the same type of word in Line 78.
10. Please use “figure 6” instead of “figure 5” in Line 139.
11. Please use “also” instead of “slao” in Line 262.
Reviewer 2 Report
The article is consice and well constructed. Although it is much enlarged it is worth publication.
Also is there any particular reason why section 2 is not included in the materials and methods?Minor ammendments should be made as follows.
In line 38 correct 7,5000 to 7,500
in line 98 delete "were"
In line 103 rewrite the first sentence
In line 109 delete "of"
In line 115 delete "that"
In line 187 delete "that"
In lines 193-194 rewrite the sentence
In line 298 delete "the"
Round 2
Reviewer 1 Report
The author has answered my questions and revised in new manuscript. There is no other question.